# *Trifolium repens*-Associated Bacteria as a Potential Tool to Facilitate Phytostabilization of Zinc and Lead Polluted Waste Heaps

**DOI:** 10.3390/plants9081002

**Published:** 2020-08-06

**Authors:** Ewa Oleńska, Valeria Imperato, Wanda Małek, Tadeusz Włostowski, Małgorzata Wójcik, Izabela Swiecicka, Jaco Vangronsveld, Sofie Thijs

**Affiliations:** 1Department of Microbiology and Biotechnology, Faculty of Biology, University of Białystok, Ciołkowskiego 1J, 15-245 Białystok, Poland; izabelas@uwb.edu.pl; 2Faculty of Sciences, Centre for Environmental Sciences, Hasselt University, Agoralaan D, B-3590 Diepenbeek, Belgium; valeria.imperato@uhasselt.be (V.I.); jaco.vangronsveld@uhasselt.be (J.V.); 3Department of Genetics and Microbiology, Institute of Biological Sciences, Faculty of Biology and Biotechnology, Maria Curie-Skłodowska University, Akademicka 19, 20-033 Lublin, Poland; wanda.malek@poczta.umcs.lublin.pl; 4Department of Evolutional and Physiological Ecology, Faculty of Biology, University of Białystok, Ciołkowskiego 1J, 15-245 Białystok, Poland; twlostow@uwb.edu.pl; 5Department of Plant Physiology and Biophysics, Institute of Biological Sciences, Faculty of Biology and Biotechnology, Maria Curie-Skłodowska University, Akademicka 19, 20-033 Lublin, Poland; m.wojcik@poczta.umcs.lublin.pl; 6Laboratory of Applied Microbiology, Faculty of Biology, University of Białystok, Ciołkowskiego 1J, 15-245 Białystok, Poland

**Keywords:** *Trifolium repens*, bacterial endophytes, metals, plant growth promotion, phytoremediation, 16S rRNA gene

## Abstract

Heavy metals in soil, as selective agents, can change the structure of plant-associated bacterial communities and their metabolic properties, leading to the selection of the most-adapted strains, which might be useful in phytoremediation. *Trifolium repens*, a heavy metal excluder, naturally occurs on metal mine waste heaps in southern Poland characterized by high total metal concentrations. The purpose of the present study was to assess the effects of toxic metals on the diversity and metabolic properties of the microbial communities in rhizospheric soil and vegetative tissues of *T. repens* growing on three 70–100-years old Zn–Pb mine waste heaps in comparison to *Trifolium*-associated bacteria from a non-polluted reference site. In total, 113 cultivable strains were isolated and used for 16S rRNA gene Sanger sequencing in order to determine their genetic affiliation and for in vitro testing of their plant growth promotion traits. Taxa richness and phenotypic diversity in communities of metalliferous origin were significantly lower (*p* < 0.0001) compared to those from the reference site. Two strains, *Bacillus megaterium* BolR EW3_A03 and *Stenotrophomonas maltophilia* BolN EW3_B03, isolated from a Zn–Pb mine waste heap which tested positive for all examined plant growth promoting traits and which showed co-tolerance to Zn, Cu, Cd, and Pb can be considered as potential facilitators of phytostabilization.

## 1. Introduction

As a result of extensive anthropogenic industrial and agricultural activities, numerous soils are polluted with metals at concentrations that are toxic to ecosystems and to human health [1,2]. Heavy metals, being inorganic compounds, do not undergo any process of biodegradation and therefore they persist in soils for decades or centuries, maintaining their potential toxicity. To reduce or even eliminate the risks of these hazardous elements in the environment different physicochemical remediation approaches (i.e., soil washing, adsorption, biosorption, precipitation, and chelation with alginates, citrates, flavonoids, and phytic acid) have been developed [3]. However, it turned out that these methods hold significant limitations leading to the disturbance of ecosystems, e.g., altering the biological and physicochemical properties of soil and/or negatively affecting indigenous soil microbial communities [4]. As an alternative to physicochemical remediation methods, a biological innovative, environmentally friendly, and cost-efficient tool, based on the usage of vascular plants to extract, sequester, and/or detoxify pollutants, generally termed as phytoremediation, has been developed [5,6,7,8].

Phytoremediation relies on various mechanisms which plants can use to deal with metal toxicity, avoiding the transfer of metals into the cells or activation of intracellular mechanisms, which provide the basis of the whole plant tolerance system against deleterious metals [9]. Plants express their tolerance to heavy metals using a strategy of (i) accumulation, i.e., extracting toxic elements from soil and accumulating them in the harvestable, above-ground plants parts; or (ii) exclusion, which relies on the ability to exclude metals from the shoots, and concentrate them on/in roots, thus reducing their further spreading due to wind or water erosion, percolation to the groundwater and the risk of additional environmental harm [5,10,11,12]. Such evolutionary acquired adaptations of plants to withstand metal toxicity can be exploited for phytoextraction or phytostabilization, respectively.

The efficiency of the restoration of metal polluted soils can be substantially enhanced by the simultaneous application of plants with microbes [13]. It has been shown that endophytes inhabiting plant tissues may significantly improve phytoremediation [6]. For each individual polluted site, the most effective plant–microbe associations must be selected. Microorganisms may exert positive effects on plants indirectly, detoxifying noxious ions using different tolerance mechanisms [14,15], and directly by influencing plant biomass production [16,17]. Plant growth promoting (PGP)-bacteria are defined as microbes able to act beneficially on plant growth by (i) synthesizing phytohormones (e.g., indolyl-3-acetic acid—IAA) or influencing their level (1-aminocyclopropane-1-carboxylic acid — ACC deaminase as an ethylene biosynthesis inhibitor); (ii) increasing the availability of nutrients, e.g., nitrogen (N_2_ fixation), phosphorus (solubilization and mineralization of phosphorus forms), and iron (by synthesising siderophores); (iii) controlling pathogens and preventing plant infections by, e.g., direct antagonism against pathogens, competition for nutrients, and activation of induced systemic resistance (ISR) in plants [5,18,19]. Indeed, some endophytes of metallicolous plants can synthesize auxins and/or siderophores and are able to supply plants with nutrients of limited availability [20]. For instance, Sánchez-López et al. [21] reported the high prevalence of endophytes isolated from *Crotolaria pumila* growing on metal mine residues with capabilities to synthesize ACC deaminase, to produce siderophores, acetoin, IAA, to solubilize phosphorus, and to fix atmospheric nitrogen.

The 70–100-year old zinc–lead (Zn–Pb) mine waste heaps, Bolesław, Bukowno and Saturn, localized in the Silesia-Kraków Upland (Poland), are post-industrial and post-mining deposits. These habitats are extremely heterogeneous, scarce in water and nutrients and with high mean total concentrations of metals up to 52,795 µg Zn, 578 µg Pb, and 605 µg cadmium (Cd) per g dry soil (see Table 1). In spite of such unfavorable conditions, metal-rich waste heaps are inhabited by vegetation covers of uneven density. Legumes (*Fabaceae*) are one of the most common families present on these sites. Legumes include several pioneer species, able to colonize marginal soils and to improve the nutritional status and organic matter content (and thus water-holding capacity) of the soils they are growing on. *Trifolium repens* (white clover) is such a species that spontaneously colonizes most Zn–Pb containing waste heaps in southern Poland [22,23].

Oleńska and Małek [23] have recently demonstrated that the high metal concentrations in the soils of these waste heaps act as a natural selection factor shaping the genetic structure of native bacterial populations. A reduction of the genomic polymorphism of *Rhizobium leguminosarum* bv. *trifolii* strains isolated from root nodules of *T. repens* growing on the old Bolesław Zn–Pb waste heap in comparison with a reference *R. leguminosarum* bv. *trifolii* population from a non-polluted grassland, and genotypes adapted to high metal conditions were revealed [15,22]. It cannot be excluded that the extreme levels of toxic metals present in metal-rich waste heaps may influence the composition of bacterial communities, their properties, and associations with plants. Bidar et al. [24] showed the underground-tissue-accumulation of toxic metals in *T. repens*, which indicates that this species is a potential excluder and possible phytostabilizer. Therefore, knowledge about the taxonomical composition of the *Trifolium* associated microbiome, particularly about the rhizospheric bacterial communities and their possible beneficial traits for the host-plant, is of great importance in a context of the use of white clover as a potential species for phytostabilization. It was reported that nodules of leguminous plants are not exclusively inhabited by rhizobia, and that they might coexist with other microbial communities including mycorrhizal fungi [25,26]. Furthermore, Sanchez-Lopez et al. [21] showed that the microbiome of vegetative plant tissues may influence the fitness of its host-plant growing on a metal polluted soil. Since there exists a huge shortage of knowledge about the microbiome of the potential phytostabilizer *T. repens*, and its probable evolutionary adaptation to long-term metal exposure, the purpose of the present study was to determine the effect of metals on the diversity and phenotypic properties of the cultivable bacterial communities inhabiting soil, nodules, roots, and leaves of *T. repens* growing on three 70–100-year old Zn–Pb waste heaps (Bolesław, Bukowno, and Saturn) in comparison to *T. repens* originating from a reference non-polluted grassland in Bolestraszyce. In order to elucidate the taxonomic position of 113 bacterial cultivable isolates obtained from calaminarian and non-polluted grasslands, 16S rRNA gene analysis was applied. Moreover, to obtain wider knowledge about potential bacterial involvement in phytoremediation, several plant growth promotion traits (synthesis of auxins, siderophores, organic acids, acetoin, ACC-deaminase activity, phosphorus solubilization, and nitrogen fixation abilities) of the studied strains as well as their metal (Zn, Cd, Cu, and Pb) tolerance, were investigated.

## 2. Materials and Methods

Bacteria were isolated from leaves, roots, nodules, and rhizosphere of white clover (*Trifolium repens* L.) growing on three (70–100 year-old) metalliferous waste heaps in Bolesław (50°17′N 19°29′E), Bukowno (50°16′N 19°28′E), and Saturn (50°17′N 19°4′E) situated in the Silesia-Kraków Upland (South Poland), and a non-polluted reference grassland area in Bolestraszyce (49°48′N 22°50′E) (Przemyskie Foothills, Southeastern Poland) in June 2017. Plant and soil samples were collected randomly from each site. Samples were taken aseptically from the upper 10 cm soil layer, using a sterilized shovel and stored in sterile plastic bags. Closed bags were kept in a temperature controlled cool box (4–8 °C), and transferred immediately to the laboratory.

### 2.1. Isolation of Plant-Associated Bacteria

#### 2.1.1. Isolation of Endophytes

Leaves, roots, and nodules of *T. repens* were subjected to a surface sterilization. After washing them in tap water, the tissue samples were immersed in a 0.1% sodium hypochlorite solution (6%–14% active chlorine EMPLURA^®^ EMD Millipore) for 10 s, and rinsed three times in sterile deionized water. The sterilization effectiveness was verified by plating 100 µL of the last rinsing water on a solid rich 869 medium, which is an appropriate medium for the detection of the diversity of plant endophytes [27]. Approximately 250 mg of plant tissue was homogenized in sterile 10 mM MgSO_4_, and subsequently 100 µL of the plant extract dilution to obtain a 10^−4^ concentration was plated on a solid 1/10 869 medium [27]. After four days of incubation at 30 °C, the single bacterial colonies showing different morphologies were plated separately on 1/10 869 plates. The pure cultures of isolated bacteria were suspended in sterile deionized water containing glycerol (15% w/v) and sodium chloride (0.85% w/v), and were stored in a freezer at −45 °C.

#### 2.1.2. Isolation of Rhizosphere Bacteria

In order to enlarge the access to the diverse soil microorganisms, a sandwich diffusion system was applied [28]. In brief, roots with tightly-adhering soil attached (rhizosphere) were suspended in 5 mL of 0.1 M phosphate-buffered saline (PBS, pH = 7.0) and incubated on an orbital shaker (160 rpm). Rhizosphere soil suspensions diluted to 10^−5^ were added to the cooled R2A medium (in w/v%, casein acid hydrolysate 0.05, yeast extract 0.05, protease peptone 0.05, dextrose 0.05, soluble starch 0.05, K_2_HPO_4_ 0.03, MgSO_4_ × 7H_2_O 0.003, sodium pyruvate 0.03, agar 1.5, pH 7.2) [29], supplied with 0.7% w/v Gelzan (Sigma-Aldrich, St. Louis, MO, USA) and 0.1% CaCl_2_ × 2H_2_O. Into such mixture a perforated plate was dipped. After solidification of agar in the plate’s holes, 47-mm diameter Nucleopore^®^ polycarbonate track-etched membranes, of a 0.05-µm pore size (Whatman, Maidstone, UK), which allow water and nutrients migration but prevent the efflux of bacteria out of the system, were placed on both sides of each perforated plate. Closed sandwich systems were buried into each soil, consistently watered with MilliQ, and incubated for three months at 20 °C. After incubation, the sandwich diffusion systems were opened under laminar flow, membranes were removed, and each plug from the agar-filled perforate plate was pushed into an opposite well in a sterile Masterblock^®^ (Greiner bio-one) filled with liquid R2A medium. Bacteria were incubated on an orbital shaker (160 rpm) at 30 °C for 4 days. Subsequently, 100 µL of resuspension was plated onto solid R2A medium, single colonies were purified, transferred to a glycerol solution (15% w/v) and stored at −45 °C.

### 2.2. Genetic Analysis

#### 2.2.1. Determination of Taxonomic Position of Bacteria Based on 16S rRNA Gene Sequence

The genomic DNA from each of the 113 isolated bacteria was extracted using the Applied Biosystems MagMAX^TM^ Total DNA Multi-Sample Ultra Kit according the manufacturer’s instructions (ThermoFisher Scientific). The DNA extraction was preceded by treatment of bacteria with a lysis buffer (lysozyme 90 mg, Tween20 54 mg, TE-buffer 90 µL, and nuclease free water 4410 µL). The DNA concentration was determined by using Nanodrop 2.0 and agarose gel electrophoresis (1% agarose in 1×TBE buffer). Amplification of a 16S rRNA gene fragment was performed using 0.2 µL 5 U/µL Fast Start High Fidelity Enzyme Blend, cooperating with specific reagents of the FastStart^TM^ HF (High Fidelity) PCR System (Sigma-Aldrich): 2.55 µL 10×concentrated with 18 mM MgCl_2_ FastStart HF Reactive Buffer, 0.5 µL 10 mM PCR grade dNTP (deoxynucleoside triphosphate) mix, and 0.5 µL 0.2 µM of each primer, diluted in 19.75 µL nuclease free water. For amplification of 16S rDNA the following set of primers [30] were used: 27F 5′-AGAGTTTGATCMTGGCTCAG-3′, and 1492R 5′-TACGGYTACCTTGTTACGACTT-3′ (Macrogen, Netherlands) in following conditions: initial denaturation at 94 °C for 5 min, 35 cycles of denaturation at 94 °C for 1 min, annealing at 52 °C for 30 sec, and extension at 72 °C for 3 min, and final extension at 72 °C for 10 min. The presence of PCR products was verified in a 1% agarose gel in 1×TBE buffer, and documented with the Gel Doc System (Invitrogen). The Sanger sequence PCR was performed by Macrogen. The 16S rRNA gene sequences obtained in the present study were quality filtered and trimmed in Geneious v4.8.5. [31]. The high quality sequences were BLAST (Basic Local Alignment Search Tool) searched for the closest relative in the NCBI (National Center for Biotechnology Information) database.

#### 2.2.2. Phylogenetic and Genotypic Analysis

Sequences of the 16S rRNA genes of the studied bacteria were analyzed with the Maximum Likelihood model using the MEGA 6.0 [32]. To compute the degree of the statistical support for branches in the phylogenetic tree, 1,000 bootstrap replicates were made. For an estimation of the relationships between individual strains of species, represented as nodes, minimum spanning trees (MSTs) were calculated; they consist of nodes connected by edges, the length of which corresponds to the distances between two individuals [33]. MSTs were calculated on the basis of nucleotide diversity according to the maximum parsimony method with the usage of the Arlequin ver. 3.5.2.2 software package [34].

#### 2.2.3. Determination of Relative Taxonomic Biodiversity in Communities

In order to determine the relative taxonomic biodiversity of studied microbial communities originating from the three metal polluted sites and the non-polluted reference site, the Shannon’s diversity index (*H’*) was used [23,35,36]. For the evaluation of the significance of the differences in taxonomical richness between studied microbial communities, their Shannon’s diversity index (*H’*) values were subjected to the non-parametric U Mann–Whitney statistical test at the significance level *p* < 0.05 with the usage of Statistica 13 software.

### 2.3. Phenotypic Traits of Isolated Bacterial Strains

#### 2.3.1. In vitro Studies Testing the Plant Growth Promoting Properties

All isolated bacterial strains were tested for their ability to (i) enhance the levels of available nutrients, i.e., phosphate solubilization, nitrogen fixation, synthesis of organic acids, and siderophores; (ii) produce plant growth regulators, i.e., IAA synthesis, and ACC deaminase activity; and (iii) prevent plants diseases, i.e., acetoin synthesis. Most of the tests were performed using colorimetric assays, and some of them were achieved by plating on selective media. The bacterial capacity to solubilize phosphorus was verified according to Pikovskaya [37] with the modifications of Nautiyal [38]. The bacterial availability to fix atmospheric nitrogen was examined according to Xie et al. [39]. The ability to synthesize organic acids was done using Alizarine Red S according to Cunnigham and Kuiack [40]. Siderophores production was tested using chrome-azurol S according to Schwyn and Neilands [41], and 284 medium [42]. The capability of bacteria to synthesize IAA was explored according to the methods described by Gordon and Weber [43] and Patten and Glick [44]. Bacterial ACC-deaminase activity was studied according to Belimov et al. [45]. Bacterial acetoin production was examined using the α-naphthol method [46].

#### 2.3.2. Metal Tolerance of Isolated Bacterial Strains

Zinc, cadmium, copper, and lead tolerance of the strains were verified using liquid 284 medium [42] supplemented with different concentrations of metal salts: 0.4 mM and 0.8 mM CdCl_2_ × 2.5H_2_O, 0.6 mM and 1.0 mM ZnSO_4_ × 7H_2_O, 0.4 mM and 0.6 mM CuSO_4_ × 5H_2_O, 0.4 and 1.0 mM Pb(NO_3_)_2_. Sterile microtitration cell culture plates (NEST Scientific), filled with 100 µL mixture of 284 medium supplemented with the appriopriate dose of metals, were inoculated with the investigated strains. After five days of incubation on a shaker (100 rpm) at 30 °C, the growth of all strains on metal supplemented media was evaluated as the absorbance measured at λ = 560 nm using a plate reader.

### 2.4. Analysis of Zinc, Lead, and Cadmium Concentration in Soils and T. repens Roots and Leaves

Heavy metal content in the Bolesław, Bukowno, and Saturn upper layer of metalliferous and reference soil as well as in plant tissues were measured in five sample replicates. To determine total Zn, Pb, and Cd concentrations in soil samples, 0.2 g of soil matrix, previously passed through a 1 mm sieve, was extracted for 15 min with 70% nitric acid (4.5 mL) and hydrofluoric acid (1.5 mL) at 180 °C (Method 3052, US EPA 1996) in a Mars 6 microwave oven (CEM Corporation, Matthews, NC, USA), whereas to measure the metal concentrations in white clover roots and leaves, tissues were treated with concentrated nitric acid (2.5 mL) and 250 µL hydrogen peroxide solution (30% w/w in H_2_O) diluted in 2.250 mL double deionized water at 150 °C (Method Plant Material, US EPA 1996) in the above mentioned microwave oven [47]. Metal analysis of these solutions was carried out by electrothermal atomic absorption spectrometry (AAS), using the Thermo iCE 3400 instrument with Zeeman correction (Thermo Electron Manufacturing Ltd., Cambridge, UK). Quality assurance procedures including the analysis of reagent blanks and standard reference material, i.e., Montana II soil (NIST^®^ 2711a, Sigma-Aldrich) and tomato leaves (NIST^®^ 1573a, Sigma-Aldrich), for soil and plant samples, respectively, were performed in parallel. The recovery of Zn, Pb, and Cd was 90%–95%. Data were expressed as mean ± SD. They were analyzed by one-way analysis of variance (ANOVA) followed by the Duncan’s multiple range test. Differences at *p* < 0.05 were considered as statistically significant.

### 2.5. Data Availability

The partial 16S rRNA gene sequences received from of all studied strains were deposited in the GenBank database under the accession numbers MN 943500–MN 943612.

## 3. Results and Discussion

### 3.1. Taxonomic Identification and Genetic Analysis of Bacteria Associated with T. repens

In order to search for novel heavy metal tolerant strains equipped with plant growth promotion properties and determining their taxonomic position, 113 endophytes from leaves, roots, and nodules of *T. repens* as well as bacteria from the *T. repens* rhizosphere, originating from three 70–100-year old Zn–Pb-rich waste heaps and a non-polluted reference grassland, were classified on the basis of 16S rRNA gene Sanger sequence analysis (Figure 1).

Amplification of the 16S rRNA genes of the different isolates with 27F and 1492R primers yielded products of about 1,300 bp. Identification of the isolates was performed by alignment of the 16S rRNA gene sequences with reference sequences available in the NCBI database using the BLAST algorithm. On this basis, the percentage of identical nucleotides in the 16S rRNA gene of tested isolates and reference bacteria, the endophytic and rhizospheric bacteria of *T. repens* were categorized as 31 species, members of 21 genera belonging to 13 families, 11 orders, and 4 phyla of the *Bacteria* domain (Figure 2). The most numerous phylum was that of the *Proteobacteria*, containing almost 45% of the isolates, represented predominantly by *Enterobacteriales* (38%), *Pseudomonadales* (31%), and *Xanthomonadales* (21%) with *Pantoea agglomerans* (29%), *Pseudomonas putida* (21%), and *Stenotrophomonas maltophilia* (100%), respectively. The phylum *Firmicutes*, comprising almost 40% of the isolates, was represented by strains belonging to the order *Bacillales* (100%), belonging predominantly to the family *Bacillaceae* (97%), represented mainly by the species *Bacillus megaterium* (39%). The phylum *Actinobacteria* (12%) was represented by bacteria of the order *Micrococcales* (83%) and *Actinomycetales* (17%). Among the *Bacteroidetes*, which comprised only 3% of the isolates, strains belonging to the order *Flavobacteriales* (70%) were dominating, with the family *Flavobacteriaceae* (67%), represented by *Flavobacterium* sp. and *Chryseobacterium lathyri*.

The rhizosphere as the “growth chamber” of distinct physicochemical conditions where *i.a.* carbon-rich molecules or biocontrol compounds influence the fitness of microbes, is the niche with the highest microbiome diversity [48], clearly higher than the endosphere microbiomes [49,50]. The rhizoplane can play a “critical gating” role [51], restricting endosphere community members to bind to root surfaces [52]. Exclusion of some specific bacterial groups might be a consequence of the host plant immune system activity [53]. The results show the occurrence of distinctive taxa specially between plant tissues and soil (Figure 3 and Appendix A). For example, *Lelliottia amnigena* was detected as a species exclusively occurring in soils. *L. amnigena* is a plant pathogen [54,55]. Therefore, its absence in plant tissues should be considered as advantageous and might be considered as a possible example of an adaptive mechanism to prevent the white clover from diseases. *B. megaterium* was detected only in plant tissues and was not found in soils. *Sphingomonas* sp. and *S. phyllosphaerae* were found exclusively in leaves, whereas *Stenotrophomonas maltophilia* was not detected in roots but was present in soils, leaves, and nodules. Taxa such as *B. megaterium*, *Sphingomonas* sp., and *S. maltophilia* show a wide range of distribution [56,57]. Their dissimilar presence in plant tissues might be due to differences in access to nutrients or specific traits of the host plant [58]. Moreover, it was shown that apart from rhizobia, which are permanent inhabitants of white clover root nodules [15,22,23], nodules are also inhabited by many other non-rhizobial bacterial taxa, e.g., *Bacillus* sp., *Pseudomonas* sp., *Stenotrophomonas maltophilia*, *Micrococcus luteus*, *Erwinia persicina*, and *Chryseobacterium lathyri* (Figure 3B). Muresu et al. [59] demonstrated that nodules of the legume plant *Hedysarum* sp. are also a niche for non-rhizobial inhabitants, e.g., *Enterobacter cloaceae*, *Enterobacter kobei*, *Escherichia vulneris*, *Pantoea agglomerans*, *Leclercia adecarboxylata*. Because co-inoculation of legumes by rhizobia and other bacterial species, e.g., *Bacillus* sp. obviously improved nodulation [60] and the availability of nitrogen [61], Sturz et al. [62] and Martínez-Hidalgo and Hirsch [25] suggested that non-pathogenic bacteria occurring sympatrically with rhizobia in nodules may influence a host plant beneficially. Such effective partnerships might be especially advantagous on the Zn–Pb polluted waste heaps. White clover nodules from the Bolesław, Saturn, and Bukowno metalliferous areas harbor many taxa similar to those from nodules of plants growing on the non-polluted reference site; they are inhabited by *Chryseobacterium* sp., *Stenotrophomonas* sp., biosurfactant producing *Bacillus* sp., and *Pseudomonas* sp. (Figure 3), which are commonly used in bioremediation.

A selective pressure of heavy metals towards the *T. repens* endophytic and rhizosphere bacterial communities from the waste heap in comparison to those from the reference area is obvious. The U Mann-Whitney test revealed a significantly higher (*p <* 0.0001) Shannon’s diversity index in the reference samples from Bolestraszyce (*H’* = 2.9532) in comparison with those from the three metal polluted waste heaps, Bolesław (*H’* = 1.8181), Bukowno (*H’* = 2.2310), and Saturn (*H’* = 1.6521). The higher diversity of the communities from the non-polluted reference site corresponds with the significantly lower Zn, Pb, and Cd concentrations in the reference soil in comparison with the waste heap soils (Table 1). In leaves of *T. repens* from the reference grassland, besides *B. megaterium,* 10 different endophytic bacterial species were identified, compared to six from Bolesław, four from Bukowno, and one from Saturn (Figure 3). Taxa like *Herbiconiux* sp., *Flavobacterium* sp., *Pedobacter suwonensis*, *Microbacterium foliorum* were found exclusively in leaves of white clover originating from the reference site in Bolestraszyce, while *Methylobacterium bullatum* was detected only in leaves from Bolesław, and *Pseudomonas viridiflava* was identified only in leaves from Bukowno (Figure 3).

In roots, *Brevibacterium frigoritolerans*, *Bacillus drentensis*, *Pseudomonas fluorescens*, *Microbacterium oleivorans*, *Pantoea agglomerans*, and *Burkholderia* sp. were found to be specific for *T. repens* from the Bolestraszyce reference site, whereas *Bacillus cereus* and *Microbacterium arborescens* were exclusive endophytes of roots from Bolesław, whereas *Curtobacterium flaccumfaciens* bacteria was found only in Bukowno. *Bacillus subtilis*, *B. simplex*, and *E. persicina* were spotted exclusively in *T. repens* root nodules from Bolestraszyce while *Chryseobacterium lathyri*, *Pseudomonas putida* were found only in nodules from Bolesław, and *Micrococcus luteus* only in nodules from Bukowno. In almost all investigated leaf and root microbial communities *Bacillus megaterium* appeared as a common taxon, and *Stenotrophomonas maltophilia* was found as a common species in soil and nodule communities of plants from metalliferous and non-metalliferous areas (Figure 3).

In order to identify the genotypes that could be potentially the most valuable for phytostabilization purposes, 16S rRNA gene sequences of all *B. megaterium* and *S. maltophilia* strains were identified and the relationship between individual strains was calculated and presented as Minimum Spanning Trees (MSTs) using the Arlequin 3.5 software package. Based on the 16S rRNA gene sequence analysis, the twenty investigated *B. megaterium* strains were assigned to seven genotypes with a sequence identity to each other of 96%–99%. Genotypes E (*f* = 0.15) and F (*f* = 0.20) consist of strains isolated from tissues of white clover growing on both the waste heaps and non-polluted reference site, genotypes A (*f* = 0.1), B (*f* = 0.15), C (*f* = 0.30), and G (*f* = 0.05) contain strains originating only from the waste heaps, whereas the genotype D (*f* = 0.05) comprises exclusively strains from the Bolestraszyce reference area (Appendix A). Strains of the genotype A, specific to the Bolesław waste heap, differed in their 16S rRNA sequence in one nucleotide from the genotype D strain, unique to the reference site. In turn, the bacteria of the genotype D differed in one nucleotide from the genotype E bacteria, which were found on all sites, and by one nucleotide from the genotype G that was specific to the Bukowno waste heap, and in one 16S rDNA nucleotide from the genotype B unique to the Bolesław waste heap. The bacteria of the genotype B differed in one nucleotide from those belonging to genotype C, which was specific to metal-polluted waste heaps, and in one nucleotide from genotype F bacteria, which occurred on all studied sites (Figure 4).

A comparative analysis of the 16S rRNA gene of *S. maltophilia* strains allowed to distinguish 10 genotypes (Appendix A). Two potential ancestor genotypes, genotypes A and H originating from calaminarian grasslands, gave three branches, and differed from each other by 12 nucleotides. The genotype A (*f* = 0.154) differed from the genotypes D (*f* = 0.154) and B (*f* = 0.077), obtained from tissues of the white clover growing on calaminarian grassland, by six and five mutations, respectively. The genotype D differed from the unique to calaminarian land genotype E (*f* = 0.077) in two mutations. The genotype B differed from the waste heap originating genotypes C (*f* = 0.077) by five nucleotides, and F (*f* = 0.077) by one nucleotide. The genotype H (*f* = 0.077) differed from the calaminarian origin genotype G (*f* = 0.154) by 7 mutations, and from the non-metalliferous origin genotype J (*f* = 0.077) by 6 nucleotides, and 9 mutations from the genotype I (*f* = 0.077). Genotypes G–J, which were obtained from soil, carried in their 16S rRNA gene more mutations than the other identified genotypes that were isolated from tissues of white clover. The relationships between the analyzed 16S rRNA genotypes of the studied *B. megaterium* and *S. maltophilia* strains originating from the waste heap and the non-polluted reference populations are presented as a MST in Figure 5.

### 3.2. In Vitro Testing of Plant Growth Promotion Traits and Heavy Metal Tolerance

In vitro testing of the 113 bacterial strains for their ability to synthesize auxins, siderophores, organic acids, acetoin, their ACC-deaminase activity, phosphorus solubilization, and nitrogen fixation capacities found that many of them exhibit plant growth promotion (PGP) properties (Appendix A). Over 11% of the isolated bacterial strains were positive for all seven PGP traits that were investigated, about 35% of the strains showed positive for six traits, 34% for five, 12% for four, and 8% for three traits. The taxonomic distribution of the plant growth promotion abilities and metal tolerance at the genus level, are presented in Appendix A.

More than 60% of the studied strains were able to synthesize auxins, siderophores, acetoin, exhibited ACC-deaminase activity and could solubilize phosphorus; 40%–58% of strains demonstrated diazotrophic activity, while 20%–37% of the isolates produced organic acids (Figure 6).

The latter trait leads to acidification of the soil and is a main mechanism for solubilization of inorganic phosphorus compounds that are unavailable to plants [63]. Phosphorus solubilization and fixation of atmospheric nitrogen by plant-associated bacteria are important properties that provide nutritional support for the host plants [64,65,66]. Synthesis of ACC-deaminase supports plant growth and development in stressful conditions by lowering the ethylene levels [67]. At low concentrations, auxins produced by bacteria can stimulate the elongation of primary roots and formation of root hairs, enhance the nutrients uptake by the host plant, and stimulate the production of root exudates [18,68,69]. Additionally, the capability of bacteria to synthesize siderophores is a highly advantageous trait, because microbes can facilitate the uptake of iron by their host. The availability of iron for the plants indeed is mostly limited [70]. Acetoine synthesis by bacteria can promote plant growth by stimulating root development and increasing the resistance of the plants against pathogens and drought stress [71].

When comparing the bacterial communities originating from the metal-polluted and non-polluted areas, differences in percentages of bacterial strains demonstrating PGP traits in vitro were observed (Figure 6). Compared to strains from the non-polluted Bolestraszyce reference site, significantly more strains of waste heap origin were able to produce siderophores, acetoin and to synthesize ACC-deaminase. Higher percentages of bacterial isolates from the Bukowno waste heap demonstrated phosphorus solubilization, nitrogen fixation, and organic acids production (96%, 59%, and 37%, respectively) in comparison to strains from the reference area (84%, 44%, and 22%, respectively), while the percentages of strains originating from the Bolesław and Saturn waste heaps and the reference site in Bolestraszyce producing auxins and organic acids, solubilizing phosphorus, and fixing nitrogen were similar (Figure 6). The occurrence of positive correlations between metal induced oxidative stress and high levels of bacteria-born phytohormones, ability to produce siderophores and other PGP traits of bacteria, and the high plant dry mass and high primary root length, is notable [72,73]. It was also earlier documented that, in the presence of metals, acetoin produced by bacteria improves the oxidative stress resistance, the ACC-deaminase activity is higher [74,75,76], IAA synthesis is higher and this can promote the formation of lateral and adventitious roots, and as a result can enhance mineral uptake by plants [18,68,69,77].

Results obtained during the present study indicate that *B. megaterium* and *S. maltophilia* were the most numerous representatives of the dominating phylum *Proteobacteria*, and were common taxa of both metallicolous and non-metallicolous bacterial communities associated with *T. repens* (Figure 3). All *B. megaterium* and *S. maltophilia* strains could solubilize phosphorus and synthesize ACC deaminase. Only one strain BolR EW3_A03, representing the genotype C, isolated from white clover roots from Bolesław, showed positive results for all seven PGP traits that were tested, 69% of the strains were positive for six of them, 13% for five, and 6% for four plant growth promoting traits. *B. megaterium* is an ubiquitous organism, which was already mentioned as an endophyte of plants [78]. Its plant growth promoting traits, e.g., phosphate solubilization, nitrogen fixation, and indole acetic acid (IAA) production were described before [79]. *Bacillus megaterium* R181 was reported as an effective stimulator of corn and wheat growth [78]. Moreover, *B. megaterium* synthesizes antibiotic-type compounds, for example a large group of lipopeptides of antagonistic activity similar to surfactins, lichenysins, itrurinA, and fengycins, which is of high importance in preventing plants against diseases, and the potential usage to improve plant growth [80,81,82].

All studied *S. maltophilia* strains were able to synthesize siderophores and auxins. Only strain BolN EW3_B03, representing the genotype E, isolated from nodules of *T. repens* growing on the Bolesław waste heap, showed positive for all tested PGP traits; 30% of *S. maltophilia* isolates were positive for six of the traits, 40% for five, and 10% for four tested PGP traits. As already mentioned in earlier studies [83,84], the PGP properties of *S. maltophilia* point to a biotechnological potential of these strains. Amongst others, it was reported that due to its production of proteolytic enzymes and through induction of pathogenesis related (PR) genes, *S. maltophilia* strain PD4560 can be used as biocontrol agent against *Ralstonia solanacearum* in sustainable agriculture [83]. Singh and Jha [84] also demonstrated that, under abiotic stress, *S. maltophilia* exhibits plant growth promoting (PGP) traits including synthesis of ACC deaminase, gibberellic acid, indole acetic acid (IAA), siderophores, and is able to solubilize inorganic phosphates.

Of the 113 isolates of the rhizosphere and endosphere of *T. repens*, about 65% showed co-tolerance to Cd, Zn, Cu and Pb (Appendix A); this group involved also *Bacillus megaterium* BolR EW3_A03 and *Stenotrophomonas maltophilia* BolN EW3_B03.

## 4. Conclusions

*T. repens* associated bacteria originating from three 70–100-year old Zn–Pb rich waste heaps, Bolesław, Saturn, and Bukowno showed some species specificity at the level of: (i) the ‘compartment’, e.g., *S. maltophilia* was isolated from soils, leaves, and nodules but not from roots, while *B. megaterium* was not found in soil but in plant tissues; and (ii) the type of site, i.e., *Methylobacterium bullatum* or *Pseudomonas viridiflava* were detected in plant tissues of waste heap origin, while *Herbiconiux* sp. or *Flavobacterium* sp. were found in tissues of white clover originating from the non-polluted reference site. Furthermore, concerning the potential plant growth promotion traits, differences were observed between strains originating from waste heaps and those from the unpolluted reference area. In general, more strains of waste heap origin showed the ability to produce siderophores, acetoin and to synthesize ACC-deaminase in comparison with those from the reference site. Among the 65% of strains that were co-tolerating Cd, Zn, Cu, and Pb, two strains, *B. megaterium* BolR EW3_A03 and *S. maltophilia* BolN EW3_B03, which showed positive for all in vitro tested plant growth promotion traits (i.e., phosphorus solubilization, nitrogen fixation, synthesis of organic acids, siderophores synthesis, IAA synthesis, ACC deaminase activity, and acetoin synthesis) can be assumed as the most promising ones for application in phytostabilization of Zn, Pb and Cd polluted areas.

## Figures and Tables

**Figure 1 plants-09-01002-f001:**
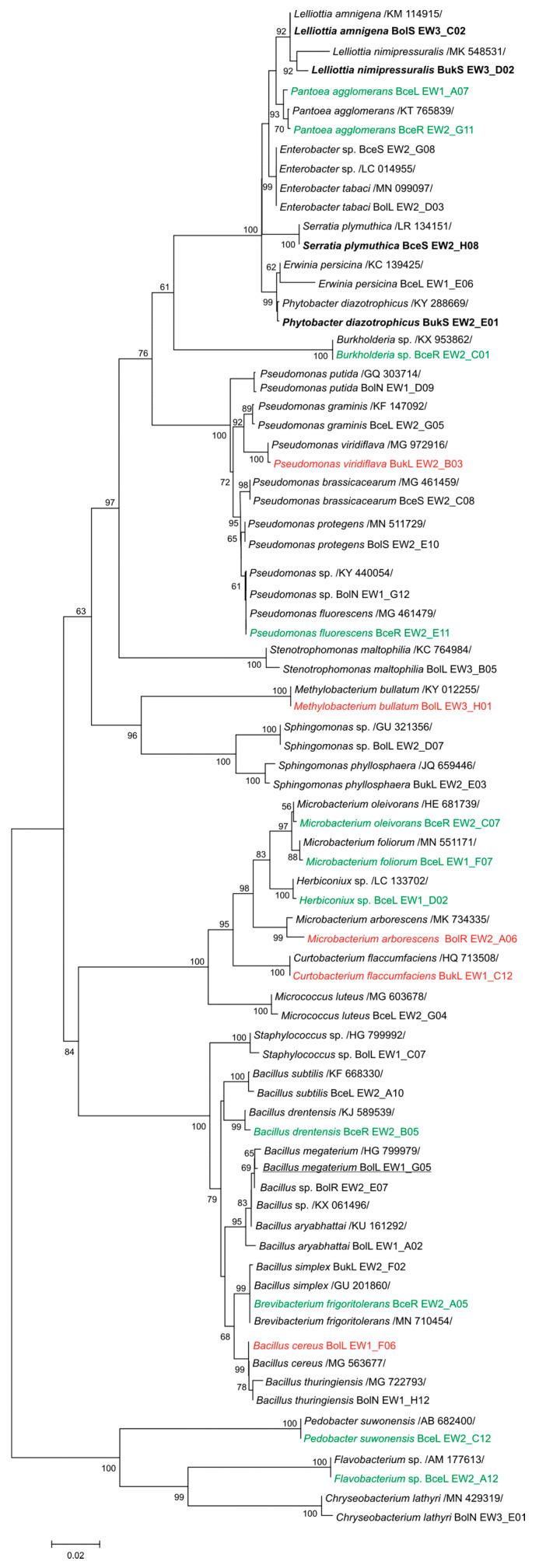
Phylogenetic Maximum Likelihood tree based on 900-bp 16S rRNA gene sequences showing the relationships of isolated bacteria and reference strains (GenBank database). Numbers at nods indicate levels of bootstrap support based on an analysis of 1,000 resampled data sets. Accession numbers of reference strains of the studied strains used as representatives are shown in parentheses. The scale bar indicates the number of substitutions per site. Names of taxons detected as characteristic to waste heap conditions are marked in red, while those to reference ones are in green. Taxon names specific to the *T. repens* rhizosphere are in bold, while those characteristic to the host-plant tissues are underlined.

**Figure 2 plants-09-01002-f002:**
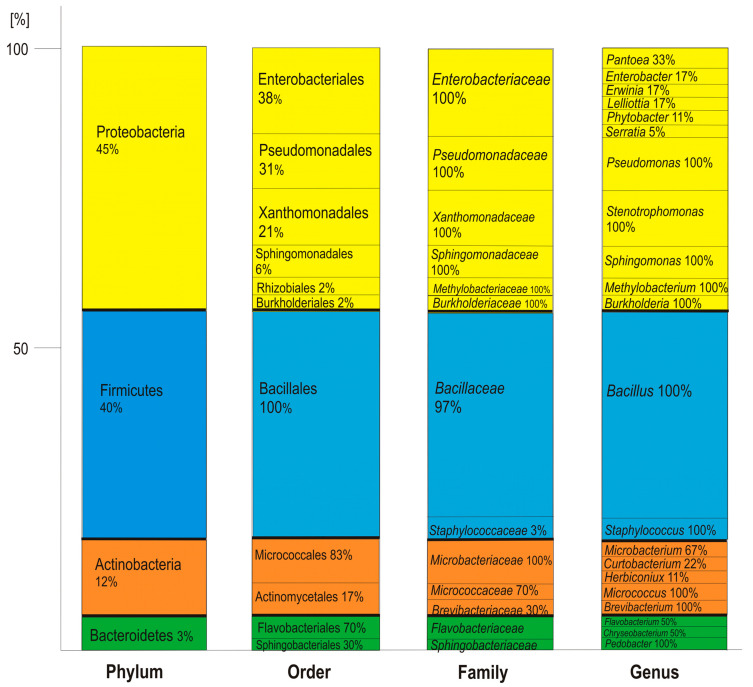
Percentages of in total 113 studied bacterial strains, identified on the basis of 16S rRNA gene sequences, in the main taxonomic groups.

**Figure 3 plants-09-01002-f003:**
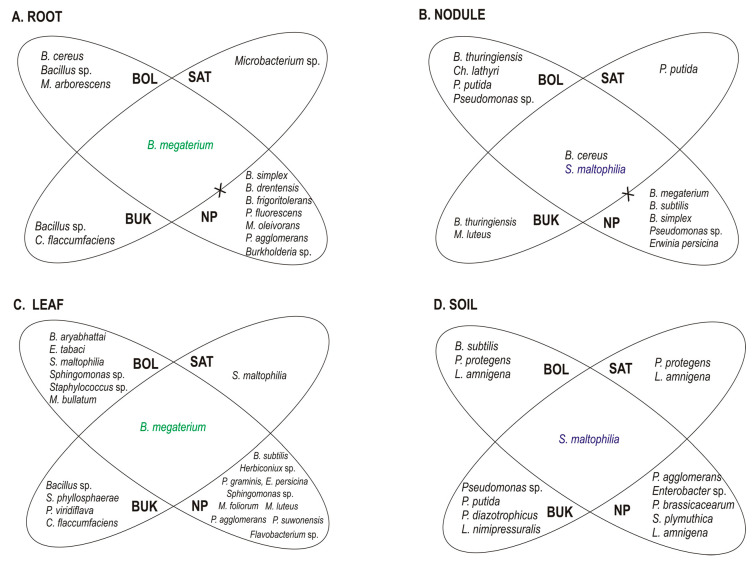
Shared and distinct bacterial taxa identified as root (**A**), nodule (**B**), and leaf (**C**) endophytes of *T. repens* and rhizosphere soil (**D**) originating from three metalliferous sites: Bolesław (BOL), Saturn (SAT), Bukowno (BUK), and the Bolestraszyce (NP) non-polluted reference site. Symbol X indices absence of common taxa.

**Figure 4 plants-09-01002-f004:**
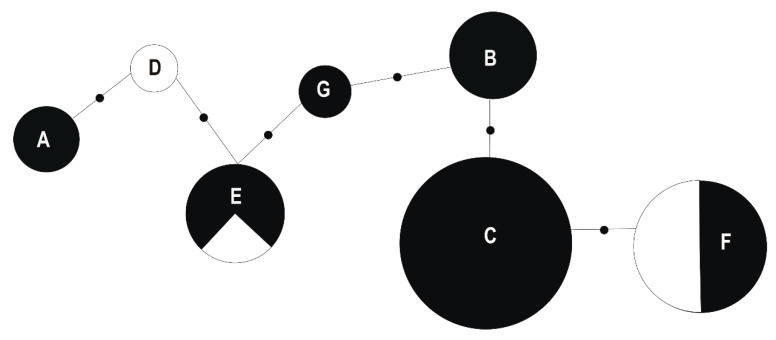
Minimum spanning tree (MST) based on a pairwise distance matrix computed on the basis of 900-bp 16S rRNA gene sequence analysis of studied genotypes (symbols A–G) of *Bacillus megaterium* strains derived from the Bolesław, Saturn, and Bukowno waste heaps (metalliferous) and Bolestraszyce non-polluted reference (non-metalliferous) area. The size of the circle corresponds to the genotype frequency. Small dots on the tree edges correspond to the number of nucleotide differences in 900-bp 16S rRNA gene between nodes representing *B. megaterium* genotypes (A–G). Dark parts of the circles correspond to the frequency of *B. megaterium* strains with a given genotype in a metalliferous area.

**Figure 5 plants-09-01002-f005:**
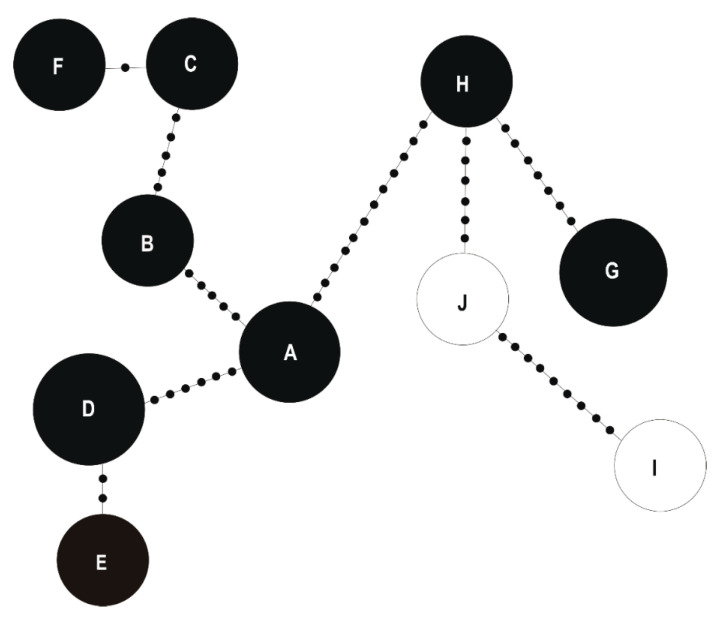
Minimum spanning tree (MST) based on a pairwise distance matrix computed on the basis of 900-bp 16S rRNA gene sequence analysis of studied genotypes (symbols A–J) of *Stenotrophomonas maltophilia* strains derived from the Bolesław, Saturn, and Bukowno waste heaps (metalliferous) and Bolestraszyce non-polluted (non-metalliferous) areas. The size of the circle corresponds to the genotype frequency in the whole sample. Small dots on the tree edges correspond to the differences in the number of different nucleotides in 900-bp 16S rRNA gene between nodes representing *S. maltophilia* genotypes (A–J). Dark parts of the circles correspond to the frequency of *S. maltophilia* strains with a given genotype in a metalliferous area.

**Figure 6 plants-09-01002-f006:**
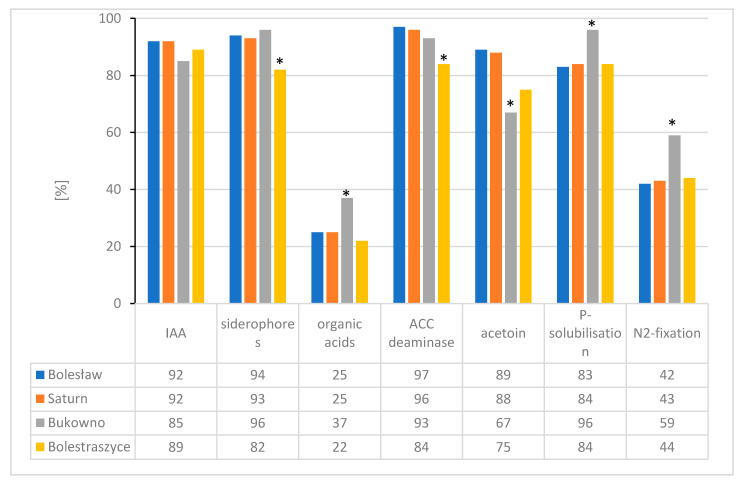
Percentages of strains isolated from rhizosphere soil and tissues of *T. repens* originating from the Bolesław, Saturn, and Bukowno metalliferous waste heaps and the Bolestraszyce non-polluted reference area demonstrating the tested in vitro plant growth promotion traits. The asterisks indicate statistically significant differences at *p* < 0.05.

**Table 1 plants-09-01002-t001:** Concentrations of total Zn, Pb, and Cd (µg g^−1^) in soils as well as in roots and leaves of white clover from the non-polluted reference area and the polluted sites in Bolesław, Bukowno, and Saturn.

	Soil			Root			Leaf		
	Zn	Pb	Cd	Zn	Pb	Cd	Zn	Pb	Cd
Bolesław	52,795 ± 1197 ^a^	578 ± 5 ^a^	605 ± 73 ^a^	609 ± 41 ^a^	243 ± 80 ^a^	288 ± 161 ^a^	329 ± 22 ^a^	7.10 ± 1.12 ^a^	1.80 ± 0.16 ^a^
Bukowno	20,159 ± 1523 ^b^	35 ± 11 ^b^	18 ± 2 ^b^	405 ± 76 ^b^	2.85 ± 0.11 ^b^	1.25 ± 0.04 ^b^	190 ± 11 ^b^	0.59 ± 0.08 ^b^	0.36 ± 0.02 ^b^
Saturn	26,016 ± 2757 ^b^	48 ± 11 ^b^	22 ± 9 ^b^	499 ± 89 ^b^	3.26 ± 0.07 ^b^	5.89 ± 0.18 ^b^	221 ± 20 ^b^	0.62 ± 0.1 ^b^	1.20 ± 0.17 ^c^
Bolestraszyce	64 ± 25 ^c^	6.87 ± 1.65 ^c^	2.56 ± 0.74 ^c^	2.7 ± 0.39 ^c^	0.45 ± 0.25 ^c^	0.51 ± 0.05 ^c^	0.41 ± 0.09 ^c^	0.01 ± 0.005 ^c^	0.08 ± 0.02 ^d^

Concentrations of Zn, Pb, and Cd are presented as mean ± SD for *n* = 5. Means in the same column marked with a different superscript letter are significantly different (*p* < 0.05) (ANOVA, Duncan’s multiple range test).

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
