# Peer review of "Trifolium repens-Associated Bacteria as a Potential Tool to Facilitate Phytostabilization of Zinc and Lead Polluted Waste Heaps"

_plants, 2020, doi:10.3390/plants9081002_

Round 1

Reviewer 1 Report

The authors study the microbiome, by genetic analysis of bacteria (113 endophytes) associated with Trifolium repens, in or out Zn-Pb mine waste-heaps of Polonia.

The introduction is very complete and fine and the methodology, based on 165 rRNA gene sequence, sound with results of possible application for phytoremediation (phytostabilization).

However, there are some flaus, weaknesses and contradictions between the initial objectives and the interpretation and discussion of the results. Therefore the content of heavy metals in soil are given, but not in plants. Moreover, the plant physiological growth promotion traits (synthesis of auxins, siderophors, organic acids, acetoin, ACC-deaminase activity, phosphorus solubilization and nitrogen abilities) not always are conclusives and are reported from a very indirecte form, so altogether the plant is very hypotheticaly considered without a direct analysis.

Author Response

Answers for Reviewer #1 comments

  1. “the content of heavy metals in soil is given, but not in plants”.

The ability of plants to accumulate the toxic metals indeed is important. We included the metal concentrations in plant roots and leaves, which is shown in the Table 1 (line 356).

  1. “…the plant physiological growth promotion traits (synthesis of auxins, siderophores, organic acids, acetoin, ACC-deaminase activity, phosphorus solubilization and nitrogen abilities) not always are conclusive and are reported from a very indirect form, so altogether the plant is very hypothetically considered without a direct analysis.”

Indeed, the reviewer is right. By in vitro testing of PGP properties, an indirect knowledge about the effects of bacterial strains on plant growth is received. We emphasized the possible role of studied strains in phytoremediation throughout the manuscript (lines: 2, 43, 109, 111, 112, 117, 124, 374, 488, 631). This problem will be subject of future study.

Reviewer 2 Report

The structure of the article need to be reviewed. The logic sequence is to have Introduction with objectives, then materials and methods, then results and discussion, and conclusion. The sequence here is messed up. You see Results and discussion comes after introduction.,.................etc

other comments include:

line ne 137: fig 1 not clear

line 162: fig 2 genus part not clear

line 216: correct numbers to the correct decimel format 

line 365: add the year of the study

line 366: add the season of plant sampling

line 369: add the depth of collected soil samples

line 376: NaOCl not correct

line 581: add year of publication

Author Response

Answers for Reviewer #2 comments

  1. The first comment of this reviewer concerns the structure of the article, suggesting applying the logic sequence as follows: introduction with objectives, materials and methods, results and discussion, and conclusion.

According to this suggestion we changed the paragraphs’ sequence, and now the order is: Introduction, Material and Methods, Results and Discussion, and Conclusions at the end of the manuscript (lines: 128-622). As a consequence of these changes, the numbers of the references in the text as well as their numbers in the References section were adapted (lines: 748-997).

  1. Other comments of reviewer 2:
  2. line 137: fig 1 not clear

According this suggestion we increased the resolution of Figure 1 (line 268 and 269).

  1. line 162: fig 2 “… genus part not clear”

We changed genus part in Figure 2 to better present genera and their percentage contribution in the families (lines 296 and 297).

  1. line 216: “… correct numbers to the correct decimal format”

We added the metal concentrations in T. repens roots and leaves (Table 1, line 356), but for reasons of readability of the table we kept the original format.

  1. line 365: add the year of the study

The year of the study was added (line 134).

  1. line 366: add the season of plant sampling

The season of sampling was included into the text (line 134).

  1. line 369: add the depth of collected soil samples

The depth of collected soil samples was added (line 135).

  1. line 376: NaOCl not correct

We changed in text NaOCl into sodium hypochlorite solution (6-14 % active chlorine EMPLURA® EMD Millipore) (lines 141-142).

  1. line 581: add year of publication

We added the year and other bibliographic data concerning the publication (lines 723-724).

Reviewer 3 Report

Manuscript ID: plants-870129

 Title:  Trifolium repens-associated bacteria as a potential tool to facilitate phytostabilization of zinc and lead polluted waste heaps

This MS written and discussed very well. Abstract is very clear and supports key objectives and achievements of current study. This MS can be published in MPDI- plants Journal, after the MINOR revision. However, there are few minor comments as:

  • Please correct the spp. As sp. throughout the MS.
  • Kindly avoid the use of “WE” line 336.
  • Quality of Figure 1 & 3 need to be improved

Author Response

Answers for Reviewer #3 comments

  1. The first suggestion of the Reviewer #3 was to change the form “spp.” into “sp.” throughout the MS.

We changed all “spp.” abbreviations into “sp.” (lines 312, 314, 318, 329-330, 349, 366, in Figure 1 lines 268 and 269, in Figure 3 lines 332 and 333).

  1. The second comment of the Reviewer #3 was to avoid of the using the word “We” line 336.

We complied with the Reviewer’s remark (line 475).

  1. The third Reviewer #3 suggestion was to improve the quality of Figure 1 and 3.

We improved the resolution of Figure 1 (lines 268 and 269) and 3 (lines 332 and 333).

Round 2

Reviewer 1 Report

The authors have taken into account the observations about the first manuscript and now the new manuscript has been improved.

Reviewer 2 Report

I think authors did the corrections i was looking for. I have no more comments